# Self-management education for hypertension, diabetes, and dyslipidemia as major risk factors for cardiovascular disease: *Insights from stakeholders' experiences and expectations*

**Nazanin Soleimani** [1], **Fatemeh Ebrahimi**[2], **Masoud Mirzaei**[3]*

1 Cardiovascular Research Institute, Cardiac Rehabilitation Research Center, Isfahan University of Medical Sciences, Isfahan, Iran, 2 School of Public Health, Bielefeld University, Bielefeld, Germany, 3 Non-Communicable Diseases Research Institute, Yazd Cardiovascular Research Centre, Shahid Sadoughi University of Medical Sciences, Yazd, Iran

* masoud_mirzaei@hotmail.com, mmirzaei@ssu.ac.ir

**Data Availability Statement:** All relevant data are within the manuscript and its Supporting information files. The interviews and FGDs were

## Abstract

### Background

Cardiovascular disease (CVD) is a leading cause of premature death, with hypertension, diabetes, and dyslipidemia as major risk factors. Effective self-management (SM) is crucial for controlling these conditions and improving quality of life. This study examines stakeholders' experiences and expectations of SM education to enhance program development.

### Methods

This study employed a qualitative grounded theory approach to explore the perspectives of three stakeholder groups: 19 patients with hypertension, type 2 diabetes, and dyslipidemia, 11 primary healthcare providers, and five provincial health policymakers and managers. Data were collected via semi-structured patient interviews and focus group discussions (FGDs) with health professionals. Coding and analysis were conducted separately using Corbin and Strauss principles with ATLAS. ti version 9.0 software.

### Results

Most patients were women (68%) aged 50–60 years (37%), with education levels from illiterate to master's degree; 32% had completed primary school. Most were housewives (52%), and 12 had multiple chronic diseases. Healthcare providers included six community health workers and five primary care physicians, with average experience of 12 and 19 years, respectively. Health policymakers and managers averaged 25 years of experience. Patient interviews and FGDs resulted in 12 and 13 subthemes, respectively, with five subthemes common to both sources. These subthemes were grouped into broader main themes, including "effective content design," "effective presentation and delivery," "characteristics

conducted in Farsi, and the coding process was performed in the same language. Subthemes and themes were translated into English for this paper. As a result, the raw data is only available in Farsi ATLAS . ti files.

**Funding:** The author(s) received no specific funding for this work.

**Competing interests:** No conflicts of interest were present in conducting this research, as confirmed by all authors.

and conditions of involved parties," and "educational needs," collectively reflect the central concept of "effective self-management education".

## Conclusion

Although the core concept and its main themes were evident and consistent across stakeholder groups, significant variations in subthemes from each stakeholder emerged. This underscores the importance of considering diverse viewpoints and highlights that, while overarching concepts may seem uniform, exploring the details of stakeholder perspectives is crucial for understanding their nuanced opinions. Effective education should integrate these insights, focusing on tailored communication, interactivity, and active monitoring.

## Background

Population aging is a global trend in developing and developed countries alike [1]. This demographic shift has led to increased chronic disease, which has now become a major public health concern [2,3]. Cardiovascular disease (CVD) is the predominant chronic disease and the leading cause of premature death in many societies [4–6]. An estimated 17.9 million people died from cardiovascular diseases in 2019, representing 32% of all global deaths[7]. Risk factors associated with CVD, such as diabetes, hypertension, and dyslipidemia, often coexist and interact, increasing the likelihood of developing cardiovascular complications [8]. In addition to contributing to cardiovascular problems, diabetes, hypertension, and dyslipidemia are associated with several other serious complications. These include microvascular damage (retinopathy, nephropathy, neuropathy) and macrovascular complications (peripheral vascular disease, cerebrovascular accidents) [9]. Diabetes, hypertension, and dyslipidemia are chronic conditions by nature, unlike other CVD risk factors [10].

People with chronic diseases typically need education on disease management to prevent complications, including effective use of medications and adopting healthy lifestyle practices [11,12]. Empowering patients to manage their conditions can significantly reduce adverse outcomes and recurrence of complications, potentially prolonging life and maintaining individual independence [13]. Studies have also shown that self-management (SM) issues are critical factors in the occurrence of stroke and premature death due to hypertension [14]. Although the benefits of managing blood glucose, hypertension, and cholesterol levels have been scientifically proven, adherence to these measures is not high [15]. This suggests significant barriers to SM [16].

In recent years, approaches to chronic disease management have shifted from the traditional patient-provider relationship to a paradigm in which people with chronic conditions play a central role in managing their healthcare through collaboration and active participation with the healthcare team [17,18].

SM encompasses an individual's daily management of chronic conditions [16,17]. In healthcare, it refers to a person's ability to cope with the symptoms, treatment, psychosocial consequences, social and physical aspects, and lifestyle changes associated with a chronic condition [19].

Studies show that providing adequate educational information on SM can reduce disease-related complications by 80%, improving quality of life [20–22].

A qualitative study by Lotte Timmermans et al. in Flanders, Belgium, involved semi-structured interviews with 16 patient-caregiver dyads using purposive sampling and inductive content analysis. The study developed the SILCQ model—Supporting, Involving, Listening,

Coordinating, and Questioning—as a conceptual framework for primary care, emphasizing its use in creating effective roadmaps and toolkits to enhance self-management support for chronic patients [23].

In another study, Marta Gorina et al. conducted a qualitative study using focus groups to identify critical factors influencing the adoption of self-care behaviors in patients with coexisting diabetes, hypertension, and hypercholesterolemia. The study involved patients, nursing professionals, and family doctors from primary care centers. Content analysis revealed that the adoption of self-management recommendations is influenced by factors related to the patients, healthcare professionals, and the healthcare system. The findings suggest that healthcare professionals should consider these broader determinants when recommending self-care [24].

Dena Schulman-Green et al. conducted a qualitative meta-synthesis of 53 studies to identify factors affecting self-management in adults with chronic illness, focusing on diabetes and cardiovascular disease. Analyzing data from diverse participants (mean age 57) across 20 countries, the study identified five key categories: Personal/Lifestyle Characteristics, Health Status, Resources, Environmental Characteristics, and Health Care System. The findings suggest that understanding these factors can improve self-management assessments and inform tailored interventions to enhance health outcomes [25].

Although valuable research has been conducted in this area, gaps still need to be addressed. A qualitative systematic review by Bruno Rushforth et al. (2016) highlights the need for strategies beyond a 'one size fits all' model for managing type 2 diabetes, emphasizing that effective improvement strategies must address organizational and individual levels [26]. Additionally, while studies underscore the importance of patient-centered interventions and consider patient preferences for education [14,27,28], qualitative research often explores these indirectly by identifying facilitators and barriers to self-management support [11,22,26,29–33]. This research project aims to fill a gap in directly addressing patient needs and preferences through comprehensive approaches that incorporate diverse stakeholders' experiences, perceptions, and expectations in self-management education programs for hypertension, diabetes mellitus, and dyslipidemia. Stakeholders in this study include patients, primary healthcare providers (primary care physicians (PCP) and community health workers (CHW)), health policy makers (HPM), and health managers (HM). The insights gathered will inform the design and implementation of tailored SM programs that address individuals' specific needs and preferences for managing chronic conditions.

## Method

This qualitative research adopts grounded theory methodology, a qualitative approach to exploring social processes and underlying structures in human interactions. This approach was considered appropriate because of the complex nature of SM education, which is influenced by multiple factors and involves diverse groups [34].

While grounded theory typically follows an inductive approach, we used the Corbin and Strauss model for self-management as a conceptual framework [35]. This model, which organizes self-management into medical, behavioral, and emotional aspects, guided our data collection and analysis. This approach helped organize and contextualize our findings within a broader understanding of self-management, ensuring that our research contributes meaningfully to the field while remaining grounded in the data.

### Target population

The target population comprises key stakeholders engaged in the SM education process for patients with hypertension, type 2 diabetes mellitus (T2DM), and dyslipidemia in primary

healthcare centers (PHC). This study involved three stakeholders: 1-patients, 2-PCP and CHW, 3-HPM, and HM.

## Participant selection

Interviews were conducted with patients selected from six PHCs, chosen for their diverse cultural, economic, and social contexts and their high client load and accessibility. These included five urban PHCs and one urban-rural center. Based on socioeconomic status, the selected PHCs were categorized from A to D. Interviews followed theoretical and purposive sampling principles in the selected centers. Potential interviewees were contacted to schedule an interview. Interviews were conducted in person or by telephone, depending on individual preference.

## Data collection

Ethical approval (IR.SSU.MEDICINE.REC.1400.331) was obtained from the Faculty of Medicine Ethics Committee, Shahid Sadoughi University of Medical Sciences. Data collection included semi-structured individual interviews with patients and focus group discussions (FGD) with health professionals. Guidelines for conducting interviews and discussions were developed based on the literature review and professional consensus and presented in S1A–S1C Table. Semi-structured patient interviews were conducted and coded in fall/winter 2022, followed by FGDs with professionals in spring/summer 2023. All interviews and FGDs were conducted in Persian (Farsi), and the coding process was also carried out in Farsi. After completing the coding, the sub-themes and themes were translated into English by the research team, including one researcher who is a native speaker of both Farsi and English. This translation process was followed by a validation session where the translated terms were discussed and verified using relevant literature and dictionaries to ensure accuracy and consistency.

## Interviews with patients

Three pilot interviews were conducted and reviewed to revise the primary interview guide. Written or recorded verbal informed consent was obtained from participants to ensure anonymity. Interviews, averaging between 30 and 90 minutes, were audio-recorded with simultaneous note-taking. Interviewing and concurrent coding continued until theoretical saturation was reached.

## Focus group discussions

Two FGDs were conducted, one with primary health care providers and one with provincial HPMs and HMs, each lasting approximately 90 minutes. The discussions were recorded, and memos were taken. We treated these FGDs as a unified component of the study focused on exploring the experiences and expectations of health professionals rather than as separate parts. As a result, we coded them together. Specifically, in Iran, PHCs consist of PCPs and CHWs, known as "Behvarz" in Farsi, who work as an integrated team. CHWs are full-time employees within the health system and are considered health care providers. They provide primary health care and work closely with community members to address the social determinants of health. Because of this collaborative approach, the roles of PCPs and CHWs overlap significantly, making their combined input critical to understanding primary health care.

The discussions were designed and conducted according to the principles outlined in "A Practical Guide to Focus-Group Research" [36].

## Coding process

Interviews and discussions were transcribed verbatim into MS Word files and analyzed using the ATLAS.ti version 9 software. The FGDs and interviews were coded separately to capture the distinct experiences and expectations of patients and health professionals as two different stakeholders. This analysis employed the Corbin and Strauss coding model, which includes three levels: open, axial, and selective. [34]. Two researchers did coding, with initial codes refined in online sessions between them. This was followed by a team session to review and finalize the codes and themes, ensuring a thorough and validated process.

The validity and reliability of the data were assessed using the criteria established by Lincoln and Guba [37] and the standards published by O'Brien et al. [38].

## Research team

This paper presents findings from the MD thesis of Nazanin Soleimani (Female). Masoud Mirzaei (Male), supervisor of the research team, holds MD, MPH, and PhD degrees in cardiovascular disease prevention and has completed a postdoctoral fellowship. Fatemeh Ebrahimi (Female) was the thesis co-supervisor and holds a PhD in Sociology. MM and FE have significant expertise in qualitative research, particularly in this field, and provided training and supervision to NS throughout the study. NS conducted interviews and FGDs under the supervision of both team members. Further details regarding contributions are provided in the author contribution section.

## Results

### Participant characteristics

**Patients characteristics.**  Nineteen patients were interviewed with informed consent, comprising 13 women (68%) and six men (32%). Participants were 40 to 80 years old, with the majority falling between 50 and 60 (37%). Participants' education levels ranged from illiterate to master's degree, with the highest proportion having completed primary school (32%). Regarding employment status, most were housewives (52%). In terms of medical conditions, nine individuals had non-insulin-dependent T2DM, nine had hypertension, and nine had dyslipidemia. Furthermore, 12 individuals had multiple chronic diseases (Tables 1 and 2).

**Primary healthcare providers' characteristics.**  An FGD with primary healthcare providers involved 11 participants. They included six CHWs, four women and two men, and five PCPs, one woman and four men, from the PHCs. On average, the CHWs had 12 years of work experience (WE) (range 8–26 years), while the PCPs had an average of 19 years of experience (range 2–27 years) (Table 3).

**Health policymakers' and health managers' characteristics.**  An FGD with HPM and HM at the provincial level involved five participants: one woman and four men. They held various professional positions, including the Yazd Chief Health officer, deputy technical director of Yazd Central Health, head of the health promotion Dept, head of the non-communicable diseases Dept, and specialist in the non-communicable diseases Dept at the Yazd Central Health. On average, participants had 25 years of WE (Table 4).

**Coding results.**  Patient interviews and health professional FGDs were coded separately using the Corbin and Strauss model according to the research objectives.

The core concept identified in both analyses was "effective chronic disease self-management education," which included four main dimensions: "effective content design," "effective presentation and delivery," "characteristics and conditions of involved parties," and "educational needs."

**Table 1. Frequency of patients' characteristics.**

| Sex | Woman | 13 (68%) |
|---|---|---|
| | Man | 6 (32%) |
| **Age** | 40–50 | 6 (32%) |
| | 50–60 | 7 (37%) |
| | 60–70 | 5 (26%) |
| | 70–80 | 1 (5%) |
| **Educational level** | Illiterate | 2 (11%) |
| | Literacy Education | 1 (5%) |
| | Elementary school | 6 (32%) |
| | Middle school diploma | 1(5%) |
| | High school diploma | 4 (21%) |
| | Associate degree | 1 (5%) |
| | Master degree | 4 (21%) |
| **Employment status** | Employed | 6 (32%) |
| | Retired | 3 (16%) |
| | Housewife | 10 (52%) |
| **Chronic diseases** | Non-insulin-dependent T2DM | 9 (47%) |
| | Insulin-dependent T2DM | 7 (37%) |
| | Hypertension | 9 (47%) |
| | Dyslipidemia | 9 (47%) |
| | Cardiovascular diseases | 2 (11%) |
| **Multiple chronic disease** | Yes | 12 (63%) |
| | No | 7 (37%) |

While the central concept and its main themes remained consistent across both stakeholder groups (patients and health professionals), there were differences in sub-themes. Some sub-themes were shared, while others were unique to each stakeholder group. It is worth mentioning that even among shared sub-themes, there were variations in the codes and descriptions.

## Patients' experiences and expectations

After coding the patient interviews, facilitators and barriers to effective SM education were extracted and categorized into four main themes.

### 1. Effective content design

*Proportionality and personalization*

Participants emphasized tailoring educational content to individual literacy levels and age. Tailoring educational interventions to personal characteristics and specific conditions, whether through design, modification, or personalization, was considered beneficial for education-centered SM to improve patient satisfaction.

Patients with limited or no literacy skills frequently cited the challenges of illiteracy as leading to feelings of hopelessness in learning, reduced self-confidence, and a perceived lack of ability to self-manage chronic disease. This issue is a significant barrier to learning and implementing SM strategies for chronic conditions.

A patient noted:

*"On television, doctors talk and explain things, but I don't understand. I am illiterate."*

*(Man-66y)*

**Table 2. Patients characteristics.**

| Patient code | PHC* | Age | Sex | Educational level | Employment status | Non-insulin-dependent T2DM | Insulin-dependent T2DM | CVD | Hypertension | Dyslipidemia | Multiple chronic disease |
|---|---|---|---|---|---|---|---|---|---|---|---|
| 1 | C-1 | 40–50 | F | High school diploma | Housewife | | * | | | | |
| 2 | C-1 | 60–70 | F | Illiterate | Housewife | | * | | * | | * |
| 3 | C-1 | 40–50 | F | Literacy Education | Housewife | * | | | | * | * |
| 4 | A | 70–80 | F | Associate degree | Retired | * | | | * | | * |
| 5 | A | 50–60 | F | Master | Retired | | | * | * | * | * |
| 6 | A | 60–70 | F | Elementary school | Housewife | | | | * | * | |
| 7 | A | 50–60 | M | Master | Employed | * | | | | | * |
| 8 | B | 60–70 | M | High school diploma | Employed | | * | | | | |
| 9 | B | 50–60 | F | High school diploma | Housewife | * | | | | | |
| 10 | B | 40–50 | F | Master | Employed | * | | | | | |
| 11 | B | 40–50 | M | Master | Employed | * | | | | | |
| 12 | D-1 | 50–60 | F | Elementary school | Housewife | * | | | | | |
| 13 | D-1 | 60–70 | M | Elementary school | Employed | | * | | * | * | * |
| 14 | D-1 | 50–60 | M | High school diploma | Employed | | * | * | * | * | * |
| 15 | D-1 | 40–50 | F | Middle school diploma | Housewife | | * | | * | * | * |
| 16 | D-2 | 50–60 | F | Elementary school | Housewife | | * | | * | * | * |
| 17 | D-2 | 40–50 | F | Elementary school | Housewife | * | | | | * | * |
| 18 | C-2 | 60–70 | M | Illiterate | Retired | | | | * | * | * |
| 19 | C-2 | 50–60 | F | Elementary school | Housewife | * | | | | | * |

* Based on socioeconomic status, selected PHCs were categorized from A to D.

Patients reported satisfaction with the tailored style of nutrition education, which focused on individual needs and characteristics. Furthermore, patients emphasized the importance of receiving personalized feedback from instructors to facilitate behavior modification and ensure successful program implementation. One of the patients remarked:

*"I got this chart. I check my blood sugar two or three times a week. I write down what I eat and when. Then, I show it to the dietitian. She looks at it and tells me stuff like, 'You didn't do*

**Table 3. Characteristics of the participating primary health care professionals.**

| | | Total | PCPs | CHWs |
|---|---|---|---|---|
| **Number** | | 11 | 5(45%) | 6(55%) |
| **Sex** | Woman | 5(45%) | 1(20% of PCPs) | 4(67% of CHWs) |
| | Man | 6(55%) | 4(80% of PCPs) | 2(33% of CHWs) |
| **Average work experience(year)** | | 15 | 19 | 12 |

Table 4. Health policymakers' and health managers' characteristics.

| Post | Education | Work experience (year) | Sex |
|---|---|---|---|
| Yazd Chief Health officer | MD-MPH-PhD (epidemiology) | 26 years | Man |
| deputy technical director of Yazd Central Health | MD-MPH | 25 years | Man |
| head of the health promotion Dept | PhD (health education and health promotion) | 22 years | Woman |
| head of the non-communicable diseases Dept | MD | 26 years | Man |
| specialist in the non-communicable diseases Dept at the Yazd Central Health | MD-MPH | 26years | Man |

*good this month,' 'This is great,' or 'Here's where you messed up.' It's like having a teacher checkin' your work and tellin' you what's what. Makes you know where you stand, ya know?"*

*(Man- 47y)*

### *Clarity and comprehensibility*

This sub-theme underscores the importance of clear instructions without oversimplifying by offering precise and detailed information. One participant stated:

*"Before, they just told us to eat good foods for diabetes, but they never said how much rice or dinner to have. This new nutrition person says we can eat like 17 grapes, which makes more sense. Before, they just said to eat less, but we didn't get what that meant. So, we'd eat two bunches of grapes, thinking it wasn't much."*

*(Man-69y)*

Another patient noted:

*"Now, in the hospital, for example, on the first day of admission, they would be like, 'Now you do the insulin yourself.' They would no longer specify, for example, 'Inject it into your leg' or 'Inject it into your hand.' They just gave the insulin unit and said, 'Go and administer it."*

*(Woman-40y).*

Another patient shared her experience with PHCs:

*"When you go there, they're like, "Hey, no sugar, no candy, no tea, no this, no that." Well, everyone knows that stuff. Even on TV, they're always saying, "Don't eat this. . . stop diabetes, don't eat that." They say it so much it's like they're just saying it to say it. They keep telling you to prevent stuff, but they don't give you any real advice."*

*(Man-47y)*

One patient stressed the importance of clear and specific information, advocating for explanations not only about the symptoms and challenges of the disease but also practical solutions for its management. Another interviewee highlighted the need to demonstrate the effectiveness of proposed teachings through statistical evidence.

### *Nonrepetitive*

Repetition of educational content may cause patients to lose motivation to seek further information. Even new and unrepeated education may fail to capture the recipient's attention because of this negative perception.

One patient noted:

*"Everywhere we went, they said the same thing. They didn't provide any additional information for me to learn something more."*

*(Woman-56y)*

### 2. Effective presentation and delivery

*Accessibility*

One of the critical points in this sub-theme was the challenges patients perceived in receiving personalized education from physicians. Patients found that physicians were reluctant to provide any education at all. Additionally, one patient perceived the educational experience as inadequate and ineffective, emphasizing the need for more precise and detailed instructions. These experiences highlight patients' difficulty accessing personalized education and the lack of a recognized resource for accurate and comprehensive education.

Some participants noted that physicians, stressed by heavy workloads and frequent visits, often have limited time to devote to education. They suggested that nutritionists might be more effective instructors in Iran.

The importance of having educational facilities close to home was emphasized by several patients. In their view, PHCs are particularly beneficial because of their neighborhood orientation and easy accessibility. Others highlighted the importance of education in known, familiar centers accessible to the general public.

*Teaching methodology and materials*

Patients identified instructors' intentions and pursuit of genuine educational goals as significant factors in accepting educational programs.

Patients' preferences for learning methods varied between traditional in-person and virtual approaches. Younger patients preferred virtual methods for convenience and accessibility, while older patients preferred in-person communication and empathy. Factors such as transportation and internet access influenced patients' preferences.

One patient proposed conducting educational sessions supervised by the PHC during community gatherings and religious ceremonies. Another interviewee recommended hosting educational sessions affiliated with the PHC in mosques or neighborhood parks.

The patients provided us with some recommendations for improved educational programs, including PHC-supervised educational events during religious ceremonies in mosques or neighborhood parks and the use of social media for education. Group-based educational programs were also positively discussed.

Interviewees expressed concerns about the promotional nature of online educational content and the resulting lack of trust in its educational goals, which hindered its uptake and use. One of the patients stated:

*"Sometimes I see things about diabetes on social media, like on Instagram, but if I look closely, it's usually advertising for something, and it's all fake because it has nothing to do with diabetes."*

*(Man -47y)*

Health officials and managers seem to have neglected the virtual space that could be valuable for health education. This lack of attention has led to the domination of the space by non-experts and potential abusers. Credible educational programs tailored to the social networks and the individuals involved are also lacking. One of the patients mentioned:

*"We used to sit together and talk about diabetes and healthy eating. It was really helpful and fun. . .It was great because we all had the same problem, so we weren't shy; we just talked openly."*

*(Woman-40y)*

In addition to using appropriate methods, incentives such as free chronic disease-specific services, including testing and medications, motivate patients to participate in educational programs.

### 3. Characteristics and conditions of involved parties

*Willingness and receptivity*

Some patients expressed interest in medical topics or a desire to learn more about their disease, which helped them accept educational programs. Many, especially those with low literacy, cited compliance and receptivity as important factors in adhering to educational initiatives.

*Beliefs and previous experiences*

Three codes were identified regarding convictions and prior educational experiences: "applying education leads to good feelings," "believing in education," and "preference for personal experience." Here are some quotes that illustrate these themes:

*"When I ate these things (which are recommended), I really saw a significant effect, like my sugar went down. That's why I continued to use them regularly."*

*(Woman-40y)*

*"After I stopped smoking, about 5 or 6 years ago, my blood pressure improved a lot. Before, when I smoked for like 35 or 36 years, it used to go up, and I felt more troubled."*

*(Man-66y)*

*" I never really listened to those lessons. I might pay attention for a week or so, but then I'd just return to my normal routine. I found my own way more comfortable. . .I figured it out on my own. Their program didn't work for me."*

*(Woman-56y)*

*Family support and companionship*

Family support is crucial in helping patients learn and improve their SM skills.
One patient shared his experience:

*"I used to go to the doctor at the health center [PHC] to learn how to inject insulin. My daughter would come with me, learn here, and then teach me at home."*

*(Man-69y)*

*Instructor's expertise and teaching skills*

Overall, patients expressed concerns about the expertise of instructors and worried that education by non-specialists or laypeople could create mistrust. They recommended training by professionals in fields such as medicine, nutrition, and traditional medicine for greater acceptance. Patients also preferred trainers from outside PHCs, citing novelty and potential for engagement.

### 4. Educational needs

*Causes and mechanisms*

This sub-theme encompasses three codes: "understanding the nature and cause of the disease," "comprehending the mechanisms of the disease and its various influencing factors," and "grasping the function and effects of medications."

As one patient pointed out:

*"I wanna understand this disease as a doctor would explain it. What is it doing to my body? How is it messing with my organs? I need details, like how high blood sugar messes with my body's system, how it all adds up to this."*

*(Woman-40y)*

### Treatment and control

This sub-theme covers four codes: "medication use," "traditional medicine," "health monitoring at home," and "Lifestyle modification."

Several patient needs emerged from the coding, including the need for education about the use of medications, especially insulin, and how to adjust its use based on lifestyle and diet. Some wanted to learn about herbal remedies and how traditional and complementary practices worked. Learning about their disease's consequences was also mentioned as a potential motivator. Patients with multiple chronic diseases said they found it difficult to differentiate symptoms and were unaware that diseases were linked, for example, blood pressure and stroke.

One patient stated:

*"My feet are always tingly and numb, like all the time. I don't know why. Is it because I don't get enough calcium? Or maybe I need to be checked for rheumatism? Or could it be because of my sugar? I have no idea."*

*(Woman-58y)*

The interviews revealed a significant need for education in recognizing and managing disease symptoms. These included diabetic foot care, a family history of genetic disorders, and knowing when to see a doctor to avoid complications.

A patient remarked:

*"I've done many tests—kidney, liver, etc., thank God they're all fine, even my eyes. But I still don't know when to see a doctor or what tests to do to make sure."*

*(Man-69y)*

Patients also mentioned the need to be educated about lifestyle changes like exercise and diet.

### Psychological and emotional management

Patients expressed the need to cope with psychological conditions resulting from their illness, including acceptance of the illness, coping with stress and anger, and strengthening their willpower.

## Health professionals' experiences and expectations

The FGDs highlighted the importance of distinguishing between disease SM education and institutionalizing cultural change. The latter, which involves embedding new health behaviors in society, is more complex and requires long-term efforts. Initiating cultural change early in

the education system is crucial, as people become less adaptable as they age. Young people can be ambassadors for health, sharing SM principles with their families and communities.

The analysis of FGDs with health professionals revealed four sub-themes related to the facilitators and barriers to effective educational support.

**1. Effective content design**

*Proportionality and personalization*

Professionals emphasized the significance of adapting content and tailoring SM education to the social and cultural context, as well as the unique conditions and characteristics of the target group (patients), including factors such as age, literacy level, language, cultural norms, and beliefs.

In this regard, the professionals expressed their opinions as follows:

*"One of the greatest learning challenges lies in tailoring education for diverse age groups. For instance, teaching complex metrics like HbA1c to an 80-year-old woman is impractical. The key is communicating in her language, a crucial aspect often overlooked by healthcare providers."*

*(PCP-2y WE)*

*"Tailoring educational content to the audience's literacy level and cultural context is critical. We must adapt our communication to the audience's literacy level in rural areas. Educational content should differ for urban, uptown, and downtown areas."*

*(HPM and HM -code2)*

Another critical issue, as one professional pointed out, is addressing the cultural barriers, stigmas, and taboos associated with chronic disease.

*Curriculum-based*

Another critical principle of content design that emerged from the FGDs analysis is the importance of a structured curriculum and lesson-based SM education. Without a structured program, educational materials can become scattered and repetitive. Therefore, it is essential to systematically categorize all educational content, define it step-by-step, and ensure adaptability based on the learner's conditions.

*Sufficiency*

Health professionals stress the importance of providing practical and understandable content with detailed explanations and concrete examples to enhance comprehension. They advise against sensational headlines and checklist-style presentations and instead advocate for content that offers actionable solutions.

One of the professionals emphasized:

*"Older people, especially those with conditions like diabetes and blood pressure, often have a diminished sense of thirst; advising them to drink 6 to 8 glasses of water daily can seem inconceivable, as they claim not to feel thirsty. To address this, I suggested keeping a bottle of water with herbal infusions nearby as a reminder, which significantly increased their water intake. This experience showed us how to increase their water consumption effectively."*

*(CHW-16y WE)*

Professionals stress the need for educational programs to highlight the benefits of SM and acknowledge the risks of uncontrolled disease. This ensures the effectiveness and relevance of

the content. They also advocate a practical approach focusing on solutions rather than causes, prioritizing cost-effectiveness and simplicity.

**2**. **Effective presentation and delivery**

*Accessibility*

Professionals emphasize that accessibility is critical to effectively presenting and delivering SM education. Priority areas include providing adequate access to education and ensuring consistent support.

*Audience attraction and retention*

SM education presentations should be engaging and compelling to ensure effective audience engagement. Offering incentives such as free visits or medication can serve as an effective strategy to motivate and engage audiences.

One of the professionals stated:

*"To ensure effective education, we provide incentives because it's human nature to want rewards. We are not only not introducing new incentives to motivate our patients at the healthcare centers, but we have also discontinued previous incentives. Given the busy schedules of physicians and healthcare providers and their inability to attend daily educational sessions, we need to rethink this approach. To enhance the credibility of the education, we should reintroduce incentives like free visits or medications. I believe incorporating such incentives can increase compliance with education by 100%."*

*(PCP-27y WE)*

As professionals emphasize, follow-up and monitoring are crucial to audience acquisition and retention. Implementing follow-up through text messages or phone calls alongside SM education is essential. One HM highlighted that such follow-up fosters effective communication between recipients and healthcare teams, creates a sense of attention, and increases engagement and compliance.

Moreover, primary healthcare providers noted that patient monitoring, regular visits, ongoing education, and group discussions facilitate the internalization of SM behaviors. Supportive monitoring also instills a sense of responsibility and accountability, encouraging patients to engage in SM education.

*Communication*

Effective communication is crucial in education, requiring active engagement between patients and their healthcare team. Thoughtful instruction and genuine support are essential in promoting effective communication.

*Teaching methodology and materials*

It is recommended that interactive methods like brainstorming be used rather than traditional non-interactive approaches such as lecturing. Effective communication begins by allowing patients to express their concerns before providing education. Building a solid relationship between the instructor and the patient and understanding the patient's daily life improves communication and helps identify positive aspects of SM. Feedback on SM performance reinforces effective communication and education. Patient feedback on educational programs helps identify areas for improvement and ensure that the participants understand and apply the educational material correctly. However, using leaflets and brochures without engaging with patients may not effectively address knowledge gaps.

Practical education in chronic disease management requires using visual teaching aids such as slides and animations, practical demonstrations, and role-playing to enhance learning. Peer education, facilitated through support groups supervised by PHCs, emerges as a highly

effective endorsed method. However, educating patients about the consequences of uncontrolled disease is a significant challenge, with inappropriate techniques potentially leading to adverse outcomes. Systematic education, on the other hand, can deliver positive results. Techniques to effectively convey such information include establishing accurate communication, offering simultaneous warnings and reassurance, personalizing education, and illustrating consequences using examples and statistics. Choosing between face-to-face and virtual education platforms presents another challenge, but experts suggest tailoring the platform to the audience's characteristics, including cultural, economic, and social factors.

One of the primary healthcare providers noted:

*"In my view, social media has significant potential for effectiveness, particularly in addressing contentious health issues like vaccination, where active engagement is prevalent. Our presence on these platforms can facilitate constructive confrontation and positive outcomes. The responsiveness of Social media, which particularly appeals to young and middle-aged demographics, enhances its effectiveness. The COVID-19 pandemic further demonstrated social media's supportive role in disseminating information."*

*(CHW-8y WE)*

Another primary healthcare provider explained her experiences as follows:

*"We gave some books to the participants of the educational courses, and when we asked if they had read them, one person said, "What do we need these books for? We can find anything we want on Google in a minute." Nowadays, the Internet and social media have a much greater impact, and no one has the patience to sit down and discuss or educate in person."*

*(CHW-26y WE)*

Virtual platforms are widely recognized by HPMs and HMs as an effective solution to the challenges of in-person health education, significantly when large participant numbers and limited physical space constrain facilities. However, usability issues, particularly for older adults, alongside Internet costs, further hinder its adoption.

**3**. **Characteristics and conditions of involved parties**

*Role model instructor*

To be effective and followed, an instructor must exemplify and model the principles he/she teaches, especially diet and exercise, to inspire healthy behaviors. Patients are less likely to follow what instructors say if they do not practice what they preach.

*Instructor's expertise and teaching skills*

Health professionals emphasize that lacking expertise and teaching skills can undermine the effectiveness of SM education. Furthermore, according to an HPM, an instructor's inability to respond effectively to recipient questions can undermine trust. Additionally, HPMs highlight the importance of instructors' ability to tailor key educational content to the patient's needs and to plan its presentation step-by-step. Finally, one PCP underscores the importance of distinguishing between essential and non-essential educational topics.

*Mentally and temporally free*

Effective instructor-learner dialogue requires a sense of calm, and calm requires appropriate psychological and working conditions, underscoring the importance of adequate support in these areas. Additionally, instructors need sufficient time to teach, which is somewhat intertwined with ensuring optimal working conditions. Similarly, learners need sufficient time to learn. Moreover, addressing learners' cognitive concerns and anxiety is crucial, as these issues

can hinder concentration, diminishing the quality and effectiveness of teaching. Hence, proactive measures must be taken to mitigate learners' cognitive concerns.

### Willingness and receptivity

According to health professionals, effective SM education hinges on the instructor's enthusiasm and commitment and a purposeful relationship between instructor and trainee. Instructors need to believe wholeheartedly in the efficacy of these educational programs in controlling and managing patients' conditions.

### 4. Educational needs

Initial and general knowledge about one's disease and related issues is crucial, as it is the first step in managing chronic diseases. As one PCP pointed out, creating an initial awareness of the disease is more important than the treatment itself. As professionals in the FGDs define, SM of chronic diseases entails conscious and purposeful behavior to improve health conditions. It involves the application of knowledge to manage the disease through behavioral and lifestyle changes. Therefore, the content of SM education should be tailored to facilitate such changes.

### Prerequisites for self-management

This sub-theme includes five different codes: "understanding of the chronic nature of the disease," "understanding of the importance of the disease," "motivation for SM and understanding the importance of education," "health literacy," and "seeking support and educating family and caregivers."

Understanding the nature of chronic disease is fundamental to effective SM. It is crucial to educate patients about the disease mechanisms and distinguishing between its symptoms and consequences. Clarifying these distinctions is essential to empowering patients to manage their condition more effectively.

One of the CHWs stated:

*"If you ask someone with high blood pressure what it means, they may associate it with symptoms like dizziness and nausea without truly understanding. My experience highlights the significance of helping people understand the true meaning of diabetes and hypertension. For instance, in a hypertension course, I aimed to explain that the heart's function is to pump blood through the vessels, and any resistance in those vessels contributes to increased blood pressure."*

*(CHW-16y WE)*

Health professionals collectively stress the significance of understanding the nature and pathogenesis of diseases, although one professional suggests that education should maintain a general level of detail without delving into specifics. This understanding extends to recognizing the chronic nature of diseases, emphasizing effective control over a complete cure.

Motivating individuals to engage in SM and educating them about its importance is critical, as it emphasizes the impact on life expectancy and quality of life. Education about the consequences of uncontrolled disease can motivate patients, but they should be approached with care to avoid additional stress or negative feedback.

Encouraging patients to take responsibility for their health is crucial for successful SM, as solely relying on healthcare providers can impede progress. Establishing the importance of SM education to patients is essential, as uncooperative participants can frustrate instructors and hinder the effectiveness of educational programs. Additionally, promoting health literacy and ensuring that patients have access to reliable sources of information and can critically evaluate educational content, especially online, is essential.

The professionals also highlight the importance of involving families in supporting individuals to improve behavior and disease control, especially for low-literacy and elderly populations.

*Components of self-management*

This sub-theme includes five codes: "lifestyle management," "medication management," "disease monitoring," "symptom management," and "psychological management."

Professionals highlight lifestyle management as a critical but often overlooked aspect of SM for chronic disease. Patients' tendency to focus solely on medication and reluctance to embrace lifestyle changes present significant challenges. Effective lifestyle management entails understanding weight control, dietary impact on the disease, recommended exercise regimens, and the effect of smoking.

Additionally, addressing traditional and complementary medicine is crucial. Excluding these practices from education can lead to patient reluctance, emphasizing the need to adapt to cultural beliefs, address misconceptions, and differentiate between complementary and pseudoscientific practices.

One of the professionals noted:

*"In chronic disease management, it's important to distinguish between complementary medicine treatments and pseudoscience. Patients who feel exhausted by conventional treatments may seek non-scientific alternatives."*

*(HPM and HM, code1)*

Experts underscored the importance of medication management in chronic disease SM. Patients should be empowered to set goals for disease control. They need to thoroughly understand their medication regimen, including names, usage conditions, dose adjustments based on activity and diet (particularly for insulin), proper insulin administration, and addressing related concerns. Managing drug interactions and timing medication use under medical supervision are also crucial.

Disease monitoring is also essential, with education covering the use of devices and tools, follow-up testing principles, and blood test interpretation.

Recognizing, managing, and seeking timely care for symptoms and side effects and preventing and detecting complications early are critical aspects of chronic disease management. Managing acute symptoms and understanding the initial steps to take during specific health crises, like chest pain or hypoglycemia/hyperglycemia, are crucial for adequate symptom control.

For example, one of the experts quoted:

*"Self-care is the ability of a person with a chronic disease to manage his/her health. For example, if you have diabetes, you should know what to do if your blood sugar gets too low or how to manage your medications if it gets too high—and where to go for help. For example, people with heart problems should know what to do if they have chest pain, where to go, and how to treat it. First aid instructions should be provided if they have a problem."*

*(PCP-25y WE)*

The last aspect of SM is psychological. Patients need to learn how to cope and live with the disease. Improving their self-esteem, reward, problem-solving, self-awareness, and stress management skills is also necessary.

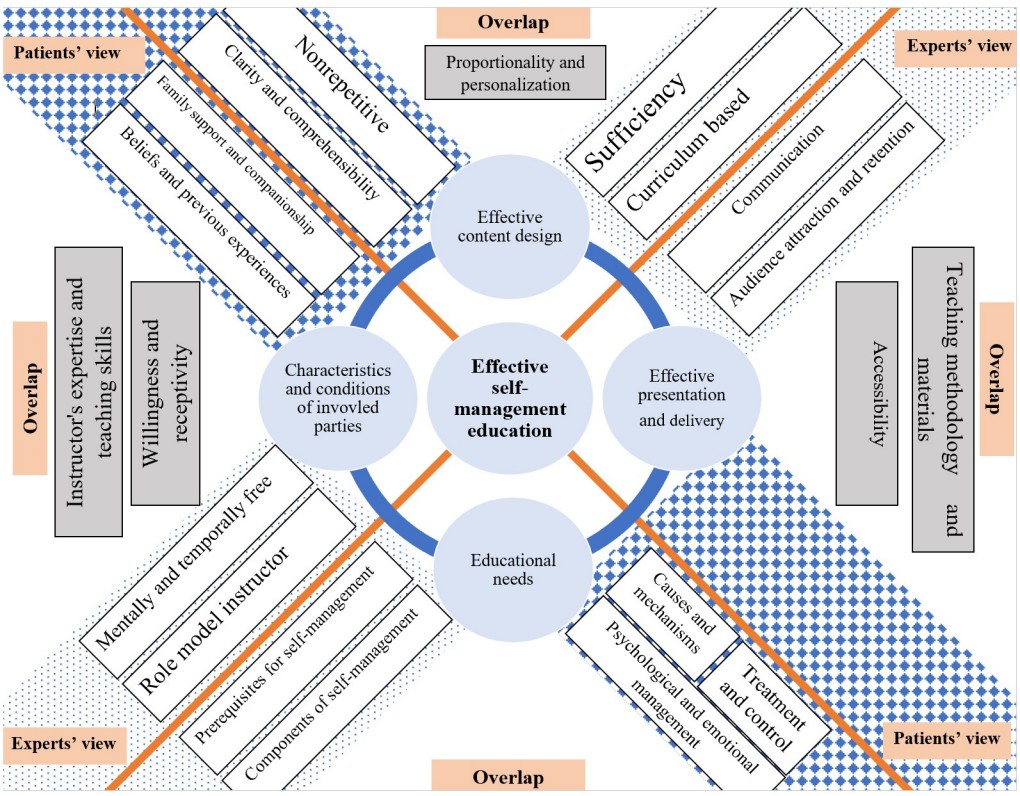

**Fig 1. Effective self-management education.**

Fig 1 provides a comprehensive visualization of effective SM education derived from coding and interpreting FGDs with health professionals and patient interviews. It illustrates the core concept, its four main themes, and corresponding sub-themes, showing their interrelationships.

## Discussion

The central concept and four main themes emerged consistently from patient interviews and health professional FGDs coding. Although some sub-themes are common to both groups, others are unique to one group, highlighting the different perspectives of patients and professionals. This underscores the need for thorough research that incorporates the insights of all stakeholders. It is important to recognize that while broad concepts may seem obvious, comprehensive research is essential to understanding the nuanced details that contribute to training effectiveness. This nuanced understanding is critical to designing and implementing optimal training strategies.

Existing studies on SM for chronic disease primarily examine facilitators, barriers, and patient SM needs, with limited qualitative research specifically addressing patient education for managing chronic conditions identified as CVD risk factors.

The findings of this study emphasize the need for SM education in chronic disease to extend beyond traditional teaching approaches. Effective communication, interaction, and indirect methods should be incorporated. Critical considerations in designing the educational approach include adhering to principles and stages conducive to behavior change, involving

instructors and learners. An initial needs assessment, informed by patients' and experts' input, should precede curriculum development. The curriculum should include concise educational modules to ensure consistent and nonrepetitive instruction.

When creating educational content, it is essential to tailor it to the audience's literacy level and socioeconomic conditions, using a personalized approach for each patient group. Content should be clear, understandable, and supported by concrete examples and statistics to facilitate patient understanding and action. Moreover, the educational environment should be adapted based on the characteristics and resources of the target group, with a choice between virtual and face-to-face platforms. Innovative and interactive methods and educational tools like group sessions, workshops, and peer-to-peer education should also be considered to enhance engagement and effectiveness.

Studies have consistently stressed the significance of patient-centered interventions and the importance of considering patient preferences for education [14,27,28]. However, qualitative research has primarily taken an indirect approach to exploring patients' needs and preferences for SM support, often understanding these needs by identifying facilitators and barriers to SM [11,22,26,29–33]. However, based on our study, we argue that patients should be considered the primary stakeholders in effective patient-centered SM education. It is essential to directly explore and understand patients' opinions to design educational programs that resonate with them. Their insights into the types of support they prefer, the features that enhance learning, and the application of teachings are critical to program design. Notably, our approach was patient-centered, and the results reflect this patient-centered perspective.

The patient-centered approach to SM support involves prioritizing the individual in treatment and self-care decisions and recognizing their ability to manage their illness and care [39,40]. In contrast to the traditional model in which patients passively receive information, the participatory approach integrates patients' needs and preferences into a personalized care plan [40,41]; it acknowledges that patients, as individuals living with the disease, have invaluable experiential knowledge that complements the expertise of healthcare providers [42–45].

Effective management of diabetes and hypertension relies heavily on SM education and patient-centered, structured behavior change counseling [46]. By fostering a collaborative and supportive relationship that promotes autonomy, this approach aims to enhance patient activation, defined as the patient's willingness and ability to participate in decision-making and manage their health independently daily [47].

Baynouna and colleagues stress the importance of health system support and patient acceptance of SM responsibility for successful SM. They note that historically, in their culture (United Arab Emirates) and others, patients often delegate their SM responsibilities to their trusted physicians, as delegation is a typical attitude for many conditions [48]. The findings of the present study confirm these observations. Moreover, they suggest that SM education encompasses more than mere instruction. Establishing effective communication and interaction and employing indirect methods are crucial aspects consistent with previous research highlighted in this study. Ineffective provider-patient communication has been associated with suboptimal diabetes care and decreased adherence to prescribed self-care practices [49]. Additionally, other studies have indicated that physicians who build trust with patients through appropriate communication, such as active listening, emotional support, and sufficient and understandable information, are more successful in ensuring patient adherence to medication regimens [50]. Miles and colleagues also support these findings by identifying "communication" as a critical factor for effective SM [51].

A meta-synthesis conducted by Schulman and colleagues also highlights "communication with healthcare providers" as an influential factor in patient SM [25]. Positive relationships

between service providers and patients facilitate SM. Through effective communication, patients can share their concerns and experience support, trust, and empathy [32,52–66].

To promote receptivity to education, individuals must be motivated and understand the importance of SM and SM education. Positive patient experiences also enhance receptivity to SM education. Nagelkerk and colleagues [54] have identified barriers such as inconvenient timing, session length, and cost implications, aligning with findings from the present study. However, while their research identified overwhelming content as a barrier, the current study highlights the need for more detailed and specific instructions, including concrete examples, as most patients and stakeholders emphasized.

Studies have underscored the challenge of educational recommendations not being tailored to individual patient needs [24,67] and the impracticality and feasibility of implementation in daily life [68]. Additionally, findings by Iglesias Urrutia and colleagues suggest that a standardized approach to decision-making for SM support interventions is inefficient [69]. The literature on decision-making emphasizes that disregarding the diversity within the patient population can have significant public health implications [70,71]. Neglecting patients' diverse preferences regarding the attributes of SM support interventions has implications for their adherence, self-efficacy, and health outcomes [72]. The study by Schaefer and colleagues suggests that individualized counseling and education may be more effective than delivering a standardized lecture on hypertension education to all patients [73].

Similarly, the study by Freire and colleagues highlights that an individual's learning experience is influenced by their economic, political, and social environment. Neglecting these contextual factors can lead to treating patients as passive learners [74].

The present study underscores the significance of tailoring educational programs to the needs of patients, taking into account factors such as literacy levels and socioeconomic conditions. It also emphasizes the need to personalize SM education initiatives, echoing findings from previous research.

This study explored the educational needs of patients regarding SM, focusing on their perspectives. Moreover, insights from health professionals obtained through FGDs delineated essential content for SM education, categorized into "prerequisites for chronic disease self-management" and "components of chronic disease self-management."

Franklin and colleagues reviewed the content of SM education and patient information needs [75]. Previous research suggests that healthcare providers are critical in providing important information for patient adherence and decision-making [76–80]. The data typically covers biomedical aspects such as blood glucose monitoring, dietary management, symptom control, risk factors, and medications [76–84]. While some patients find the information provided sufficient and valuable [77,82], others express a need for assistance applying the knowledge to their circumstances [62,79,82].

The meta-synthesis conducted by Schulman and colleagues highlighted the significance of understanding disease processes, the role of medications, and treatment plans for successful SM [85]. Crucially, individuals stressed the importance of applying SM knowledge to daily life; lack of this understanding posed a challenge to their efforts [31,32,53,54,56–58,66,86–96].

In a study by Mshunqane et al., patients demonstrated significant knowledge gaps regarding the role of lifestyle modifications in preventing diabetes complications and understanding the chronic nature of the disease [97]. Similarly, three qualitative studies in rural South African provinces found a severe lack of understanding of the causes of T2DM and hypertension, which negatively affected adherence and control [98–100].

These findings are consistent with the present study that identified the educational needs of the patients. Moreover, similar to some of the referenced studies, the present research underscores the importance of the practical applicability of educational content.

Some of the SM needs identified by patients and discussed by stakeholders in this study echo findings from previous research that examined barriers and facilitators to SM. Barriers identified include a lack of awareness of proper nutrition and preventive measures for disease complications [24,101,102] and insufficient self-motivation [16,24]. Conversely, studies have identified several factors that facilitate effective SM, including an understanding of disease mechanisms and pathology [103], motivation [16,24], recognition that effort and behavior change yield results, and acceptance of self-efficacy and one's role in disease management [103].

Numerous studies underscore the significant role of the family in chronic disease SM, serving as both facilitators and motivators [24,101,104–107]. Emphasizing the importance of family involvement, particularly in diabetes care programs, research supports educating family members about healthy lifestyle choices [108]. For instance, Zalek et al. found that training primary family members in diabetes self-care principles was a motivational factor in the SM process [109].

Several studies have demonstrated the effectiveness of family-centered education and empowerment models in promoting self-care behaviors. Family empowerment initiatives have been shown to improve knowledge, attitudes, and performance, increase self-care skills, improve quality of care, accelerate patient recovery, and reduce disease complications [110]. While the current study focused primarily on factors influencing chronic disease SM education rather than SM, the findings underscore the importance of educating families and caregivers to meet their educational needs. Additionally, patient interviews highlighted the pivotal role of family support in facilitating the learning process of chronic disease SM, a finding that has not been consistently observed in similar studies.

The current study provides a comprehensive set of findings based on patient and health professional opinions regarding appropriate platforms, methods, and education techniques. Most people prefer to receive SM education in a face-to-face setting, and although some evidence supports the effectiveness of certain SM education delivered online or by telephone, most individuals prefer receiving such education in face-to-face settings [69]. The present study's findings show diverse opinions among patients on this issue, probably due to differences in social conditions, literacy levels, age, and profession. Further quantitative and qualitative studies are needed to understand patient preferences better.

Individual education may be more beneficial for some patients, while others may prefer group sessions, according to a study by Cramm and colleagues [111]. Highly motivated and well-informed patients may require less education, while others may need to become better informed and encouraged. Therefore, SM education should address these different needs. Others have found that people who have diabetes prefer group sessions where they can share their experiences [112,113]. Group-based programs provide a peer-supportive environment where participants share experiences and learn tailored SM strategies [114]. Qualitative research reviews highlight the importance of social support, bonding, and shared learning in groups [115]. However, studies focusing on group facilitators emphasize providing evidence-based education and behavior change guidelines rather than facilitating group interactions and peer support [116–118]. The results of this study are consistent with the findings above and highlight the importance of group education, particularly peer-led and peer-supported groups, for both patients and providers.

Based on our study, patients emphasized the importance of understanding their chronic disease consequences and outcomes as a motivator for SM, which professionals also recognize. Findings highlight the importance of tailoring educational strategies to address each patient's disease consequences and outcomes systematically. Conversely, previous research suggests that providers often use consistent messages and persuasive techniques, such as threats and scare tactics, to promote patient compliance, often using disease-centered approaches [79,80,82].

Beyond the use of anxiety-provoking strategies, the results of this research emphasize the importance of providing information about the consequences and side effects of illness. While some patients found that providers' use of scare tactics resulted in more precise management of their disease [82], others experienced more anxiety and distress. To avoid anxiety, some patients preferred limited knowledge about disease progression/outcomes [79,82].

## Strength and limitations

Using a patient-centered approach, this study directly explores patients' perspectives on SM education while integrating the insights of health professionals to develop informed solutions. This study's methodological approach is a key strength that differentiates it from other studies. Following Corbin and Strauss, the study used purposive sampling to achieve theoretical saturation. Factors such as medical conditions, demographics, and socioeconomic background were carefully considered to enhance data comparability. The inherent limitations of qualitative research regarding generalizability should be acknowledged, although the findings may provide insights applicable to similar cultural contexts. Further research using larger sample sizes and mixed methods (quantitative and qualitative) is recommended to improve the design and conduct of research.

The grounded theory methodology used in this study ensures systematic and comprehensive data organization and analysis. However, it also introduces potential bias and subjectivity inherent in the researcher's interpretation. Although including FGDs in addition to semi-structured interviews enriches the study's findings, there are limitations to consider. These include limited sessions and participants due to time constraints, scheduling conflicts, and minimal involvement of provincial HPMs and HMs. Furthermore, different perspectives within groups, influenced by hierarchy or prior knowledge, may have influenced responses. Despite these limitations, the study findings provide valuable insights for designing models for educating those with chronic disease. Given the need for improved chronic disease SM education and the current lack of evidence, the findings are relevant to Iran and similar low- and middle-income contexts. However, their generalizability is limited due to the qualitative nature of the research.

## Supporting information

**S1 Checklist. COREQ checklist (consolidated criteria for reporting qualitative research).** (DOCX)

**S1 Table. Interview and focus group discussion guides.** (S1A Table: Semi-Structured Interview Guide for Patients, S1B Table: Focus group discussion Guide for Primary healthcare providers, S1C Table: Focus group discussion Guide for provincial health policymakers and health managers). (DOCX)

**S1 Appendix. Raw data (Atlas.ti file including verbatim transcription of interviews and focus group discussions in Farsi).** (RAR)

## Acknowledgments

We extend our sincere gratitude to the HPMs and HMs of Yazd, as well as to Dr. Mohsen Mirzaei, Yazd Chief Health Officer, Dr. Masoud Sharifi, deputy technical director of Yazd Central Health, Dr. Fatemeh Zare, head of the health promotion Dept, Dr. Mirmohammad Ali Evazi

Nezhad, head of the non-communicable diseases Dept, and Dr. Mohammadreza Sadeghian, a specialist in the non-communicable diseases Dept at the Yazd Central Health, for their invaluable cooperation and assistance in implementing this research and participating in the FGD.

We also express our appreciation to all the primary healthcare providers who participated in the FGD or assisted in identifying patients for interviews. Due to the large number of people, we are unable to name each individual here, but their contribution is sincerely acknowledged. Additionally, we extend our heartfelt thanks to all the patients who participated in the interviews.

Special thanks are due to Dr. Mahdieh Ghanbari for her conscientious assistance in conducting the qualitative research, valuable comments, and facilitation of FGDs.

## Author Contributions

**Conceptualization:** Nazanin Soleimani, Fatemeh Ebrahimi, Masoud Mirzaei.

**Data curation:** Nazanin Soleimani.

**Formal analysis:** Nazanin Soleimani, Fatemeh Ebrahimi.

**Investigation:** Nazanin Soleimani.

**Methodology:** Nazanin Soleimani, Fatemeh Ebrahimi, Masoud Mirzaei.

**Project administration:** Nazanin Soleimani.

**Supervision:** Fatemeh Ebrahimi, Masoud Mirzaei.

**Validation:** Masoud Mirzaei.

**Writing – original draft:** Nazanin Soleimani.

**Writing – review & editing:** Nazanin Soleimani, Fatemeh Ebrahimi, Masoud Mirzaei.

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
