## [Decision Letter · Decision Letter 0]

6 Aug 2024

PONE-D-24-14174Self-management education for hypertension, diabetes, and dyslipidemia as major risk factors for cardiovascular disease: Insights from stakeholders ' experiences and expectationsPLOS ONE

Dear Dr. mirzaei,

Thank you for submitting your manuscript to PLOS ONE. After careful consideration, we feel that it has merit but does not fully meet PLOS ONE’s publication criteria as it currently stands. Therefore, we invite you to submit a revised version of the manuscript that addresses the points raised during the review process.

We look forward to receiving your revised manuscript.

Kind regards,

Adedayo Ajidahun

Academic Editor

PLOS ONE

Journal Requirements:

Reviewers' comments:

Reviewer's Responses to Questions

**Comments to the Author**

1. Is the manuscript technically sound, and do the data support the conclusions?

Reviewer #1: Yes

2. Has the statistical analysis been performed appropriately and rigorously? 

Reviewer #1: N/A

3. Have the authors made all data underlying the findings in their manuscript fully available?

Reviewer #1: No

4. Is the manuscript presented in an intelligible fashion and written in standard English?

Reviewer #1: Yes

5. Review Comments to the Author

Reviewer #1: There is a need for more and better tools to assist patients in disease self-management. Therefore, the paper has good value for the development of those tools. Due to the overlap in health education and behavioral goals, the authors reasonably address hypertension, diabetes and hyperlipidemia in this single study. Overall, the project and this manuscript appear to be reasonably well done.

1. The second sentence of the abstract results should clarify that it concerns the patients group. It also requires rephrasing since it claims that most completed primary school but gives the percentage of just 32%.

2. The abstract results present the subthemes without first presenting the themes.

3. The abstract conclusion should be improved. Currently it has two general statements that were known to be true without this new study. Instead, summarize the findings, particularly that which is new.

4. The manuscript refers to supplementary materials, but I could not find them on the PONE website.

5. Introduction – the second sentence refers to deaths per year without stating the country or countries.

6. The introduction provides no review of prior literature on this topic, no summary of missing information, no mention of specific goals of the study.

7. The Methods do not describe a conceptual model for understanding the role of education in health behavior change or disease self-management.

8. The Methods does not provide information on the development and specific questions in the patient interview. Such information is also lacking for the questions used in the focus groups. Perhaps some of this was in the Supplemental files that I cannot find?

9. The data are said to be available but no information about accessing them is provided.

10. The authors indicate that their sample sizes allowed them to reach saturation. This may be true of the 19 patients, but I am not so sure since the patients had different CV disorders.

11. I doubt the authors reach saturation with the primary providers since it included just 6 community health workers and 5 primary care providers. These two types of professionals typically have very different education and roles. The 5 health administrators are probably too few to report.

12. Related to the above, in the US CHWs are not considered to be "health care providers." Please explain their respective roles in Iran and justify combining them in a single focus group.

13. On lines 143-145, the authors mention use of “specific research objectives” but these are not listed or explained.

14. Methods -- Were all the interviews in Persian/Farsi? Describe how translation to English was done and whether this occurred. before or after coding?

15. In qualitative research, coding is done by two individuals. This study used just one, which appears to be a weakness. Please address.

16. Results – Table 2 is redundant with Table 1 and can be removed.

17. The figure is creative, and I like it. However, is it true that the bottom right is blank (no patient options on “Effective presentation and delivery”)?

18. The Discussion paragraph on limitations should also mention limited generalizability outside of Iran and countries with similar culture and medical systems.

6. PLOS authors have the option to publish the peer review history of their article (what does this mean?). If published, this will include your full peer review and any attached files.

Reviewer #1: **Yes: **Matthew F. Muldoon, MD, MPH

---

## [Author Response · Author response to Decision Letter 0]

28 Aug 2024

We want to express our gratitude for your time and effort in providing valuable feedback on our paper. We have thoroughly reviewed your comments and suggestions and made revisions to address them. 

The response to reviewer file presents a detailed response to comments and concerns. We have also used highlights to describe the changes made in the main manuscript in response to your feedback.

---

## [Decision Letter · Decision Letter 1]

11 Sep 2024

Self-management education for hypertension, diabetes, and dyslipidemia as major risk factors for cardiovascular disease: Insights from stakeholders ' experiences and expectations

PONE-D-24-14174R1

Dear Dr. mirzaei,

We’re pleased to inform you that your manuscript has been judged scientifically suitable for publication and will be formally accepted for publication once it meets all outstanding technical requirements.

Kind regards,

Adedayo Ajidahun

Academic Editor

PLOS ONE

Additional Editor Comments (optional):

Reviewers' comments:

Reviewer's Responses to Questions

**Comments to the Author**

1. If the authors have adequately addressed your comments raised in a previous round of review and you feel that this manuscript is now acceptable for publication, you may indicate that here to bypass the “Comments to the Author” section, enter your conflict of interest statement in the “Confidential to Editor” section, and submit your "Accept" recommendation.

Reviewer #1: All comments have been addressed

2. Is the manuscript technically sound, and do the data support the conclusions?

Reviewer #1: Yes

3. Has the statistical analysis been performed appropriately and rigorously? 

Reviewer #1: Yes

4. Have the authors made all data underlying the findings in their manuscript fully available?

Reviewer #1: Yes

5. Is the manuscript presented in an intelligible fashion and written in standard English?

Reviewer #1: Yes

6. Review Comments to the Author

Reviewer #1: I commend the authors for their responses to all of my questions and suggestions.

7. PLOS authors have the option to publish the peer review history of their article (what does this mean?). If published, this will include your full peer review and any attached files.

Reviewer #1: **Yes: **Matthew F. Muldoon

---

## [Editor Report · Acceptance letter]

16 Sep 2024

PONE-D-24-14174R1 

PLOS ONE

Dear Dr. Mirzaei, 

I'm pleased to inform you that your manuscript has been deemed suitable for publication in PLOS ONE. Congratulations! Your manuscript is now being handed over to our production team.

Kind regards, 

on behalf of

Dr. Adedayo Ajidahun 

Academic Editor

PLOS ONE